# Therapeutic Potential of *Scolopendra subspinipes*: A Comprehensive Scoping Review of Its Bioactive Compounds, Preclinical Pharmacology, and Clinical Applications

**DOI:** 10.3390/toxins17050229

**Published:** 2025-05-05

**Authors:** Ye-Seul Lee, Yoon Jae Lee, In-Hyuk Ha

**Affiliations:** Jaseng Spine and Joint Research Institute, Jaseng Medical Foundation, Seoul 06110, Republic of Korea; yeseul.j.lee@gmail.com (Y.-S.L.); goodsmile8119@gmail.com (Y.J.L.)

**Keywords:** *Scolopendra subspinipes*, scolopendrasin IX, SsmTX-I, LBLP, clinical practice guidelines

## Abstract

*Scolopendra subspinipes*, commonly known as the Chinese red-headed centipede, has been utilized in traditional East Asian medicine for centuries to treat conditions such as chronic pain, inflammation, convulsions, and infections. Recent pharmacological investigations have uncovered a wide array of bioactive molecules—including peptides, alkaloids, and polysaccharide–protein complexes—from both venom and whole-body extracts. This review synthesizes findings from 45 in vitro, in vivo, and clinical studies investigating the pharmacological effects of venom-derived and whole-body-derived compounds from *S. subspinipes* across multiple domains, including analgesic, anti-inflammatory, antimicrobial, antifungal, antioxidant, antitumor, antithrombotic, anti-fibrotic, and neuroprotective activities, along with a brief scoping review of clinical practice guidelines. Key venom-derived compounds such as the peptide SsmTX-I, immunomodulatory antimicrobial peptide scolopendrasin IX, and antitumor peptide scolopentide exhibit strong mechanistic rationale and preclinical efficacy, positioning them as lead candidates for clinical development. Compounds derived from whole-body extracts, including alkaloids and polysaccharide–protein complexes, also demonstrate promising therapeutic potential. Mechanistic studies suggest these compounds operate via distinct pathways—such as ion-channel inhibition, NF-κB suppression, and apoptosis induction—offering potential advantages over existing therapies. However, current evidence remains primarily preclinical, and challenges such as extract variability, immunogenicity, and lack of standardized dosing must be addressed. Future research should prioritize isolation and structural optimization of key peptides, standardized formulation development, toxicological profiling, and early-phase human trials. The integration of traditional knowledge and modern pharmacological insights underscores the potential of venom- and whole-body-derived *S. subspinipes* agents to enrich the drug discovery, particularly for conditions with unmet therapeutic needs.

## 1. Introduction

*Scolopendra subspinipes* (*S. subspinipes*), commonly known as the Chinese red-headed centipede, has a long history of use in traditional East Asian medicine for ailments ranging from chronic pain and arthritis to convulsions and other inflammatory conditions [1]. Modern scientific interest in this species has grown as researchers have isolated a diverse array of bioactive compounds from its body and venom, including peptides, proteins, polysaccharide–protein complexes, and small-molecule alkaloids. These compounds have demonstrated a remarkably broad spectrum of pharmacological activities in preclinical studies. Notably, extracts from the whole-body of *S. subspinipes* as well as its venom have been reported to exhibit analgesic, antifungal, anti-inflammatory, antimicrobial, antitumor, antithrombotic, antioxidant, anti-melanogenic, and neuroprotective effects in various experimental models. Such multi-domain therapeutic potential makes *S. subspinipes* an intriguing subject for drug discovery and integrative medicine [1].

Considerable progress has been achieved in the scientific validation and standardization of Traditional, Complementary, and Integrative Medicine (TCIM), especially in Korea. Over 70 new or revised clinical practice guidelines (CPGs) have been published, reflecting extensive efforts to ensure evidence-based practice and improve the safety and overall quality of care, acupuncture injection therapy, or pharmacopuncture. These guidelines address a diverse spectrum of conditions, from musculoskeletal disorders to mental health issues, and are intended not only to enhance patient outcomes but also to facilitate the integration of traditional interventions within the broader healthcare system on a robust scientific basis [2]. In this context, the clinical application of compounds derived from *S. subspinipes* can be better understood and implemented in clinical practice, supported by clinically validated evidence.

Despite centuries of empirical use, the precise mechanisms underlying the centipede’s medicinal effects have only begun to be elucidated through scientific research. This article provides a structured review of the current evidence for bioactivities derived primarily from *S. subspinipes* venom components and whole-body extracts, including specific peptides and other molecular entities identified in recent studies. We organize the findings by therapeutic domain, summarizing key studies and their outcomes. We discuss the integrated significance of these findings, and the limitations and challenges that remain. Furthermore, we provide a brief scoping review of how *S. subspinipes* is recommended in the CPGs focusing on Korean Medicine to offer an up-to-date applicability of compounds of *S. subspinipes* in clinical settings, supported by clinically validated evidence. By consolidating these insights, we aim to highlight how the centipede’s diverse bioactive components could inspire novel treatments or adjunct therapies in modern medicine.

## 2. Results

### 2.1. Literature Search and Selection Process of Preclinical and Clinical Studies

A total of 123 potentially relevant studies were initially identified. Fourteen duplicates were removed, and 32 were excluded after screening titles and abstracts. Upon full-text review, 42 studies were further excluded because they were either not in English, reviews, overviews, or editorials, studies focusing on the biology and biodiversity of centipedes themselves or biological analysis of its venoms, or studies on accidental centipede bites. Consequently, 45 studies were included in the final review (Figure 1, Table 1). The PRISMA flow chart documents each step of the selection process, and Appendix A provides further details on the studies included.

### 2.2. Scoping Review on Therapeutic Activities of S. subspinipes from the Literature

#### 2.2.1. Analgesic Effects

Animal studies imply the analgesic effect of *S. subspinipes*-based therapies in arthritic pain conditions. In a rheumatoid arthritis rat model, the traditional Chinese formula Zhi Jing San alleviated joint pain, suppressed inflammation, and mitigated bone erosion by inhibiting RANKL/NF-κB-mediated osteoclastogenesis and reducing proinflammatory cytokine release [3]. In rodent acute pain assays, a venom-derived peptide (SsmTX-I) from *S. subspinipes* served as a potent Kv2.1 ion-channel blocker that attenuated pain behaviors [4]. Its analgesic efficacy matched that of morphine without provoking tolerance or other opioid-related side effects [4].

Likewise, in a mouse model of oxaliplatin-induced peripheral neuropathy, centipede pharmacopuncture (subcutaneous injection of 0.5% *S. subspinipes* extract 20 μL at the ST36 acupuncture point) markedly reduced tactile allodynia. The extract was obtained from dried *S. subspinipes* with head and legs removed, through the aqueous extraction process [5]. The effect required local neural blockade and spinal α2-adrenergic receptor activation [5]. Notably, centipede treatment showed analgesia comparable to high-dose clonidine but without adverse effects; likewise, co-administration of low-dose clonidine and centipede extract yielded synergistic pain relief with no observable adverse effects [5].

#### 2.2.2. Anti-Inflammatory Effects

Traditional medicine has long employed *S. subspinipes* for its anti-inflammatory properties, particularly in conditions such as rheumatoid arthritis [3]. Contemporary research shows that centipede extracts and centipede-containing formulations modulate inflammatory responses [6]. Mechanistically, these therapies downregulate inflammatory pathways, including inhibition of NF-κB activation (preventing IκBα degradation and p65 nuclear translocation) [7], and lowering expression levels of pro-inflammatory cytokines (TNF-α, IL-1β, IL-6) and enzymes (COX-2, iNOS) [8,9].

Certain peptides directly influenced immune cells. For example, a novel antimicrobial peptide, scolopendrasin X, originally isolated and purified biochemically from the venom of *Scolopendra subspinipes* and subsequently chemically synthesized following sequence identification, activated neutrophils via formyl peptide receptor 2 (FPR2) but diminished lipopolysaccharide-induced TNF-α and IL-6 release [10]. In autoimmune arthritis models, Zhi Jing San significantly mitigated collagen-induced arthritis by inhibiting NF-κB-mediated inflammatory cascades and lowering key cytokine levels [11]. The Soufeng sanjie formula (centipede, scorpion, and additional herbs) likewise ameliorated arthritis by decreasing IL-6, TNF-α, and IL-17 and reestablishing the Th17/Treg balance [11]. Additional evidence indicates that *S. subspinipes* extract, obtained from dried *S. subspinipes* with head and legs removed and through the aqueous extraction process as mentioned in the previous study [5], can limit neuroinflammation in nerve injury models by preventing IκBα degradation and downregulating iNOS and COX-2, thereby expediting functional recovery [12].

#### 2.2.3. Antimicrobial Effects

*S. subspinipes* extract with antimicrobial compounds include crude hemolymph and tissue extracts which exhibit potent activity against both Gram-positive and Gram-negative bacteria [13]. Chemical and transcriptomic analyses have revealed antimicrobial peptides (AMPs) and small-molecule metabolites (e.g., jineol) that contribute to its survival in microbe-rich habitats [13,14]. Many isolated AMPs display powerful bactericidal efficacy with minimal toxicity to mammalian cells [15,16,17,18,19].

Scolopin 1, a cationic AMP isolated from centipede venom, shows extensive activity against bacteria, fungi, and even tumor cells, and can be produced recombinantly [20]. Mechanistically, these peptides can disrupt bacterial membranes through pore formation or impair intracellular processes such as protein synthesis [21,22]. Some AMPs, including scolopendrasins III, V, VII, and X, also act as potent immunomodulators by recruiting neutrophils and macrophages (via FPR2 signaling) and curbing excessive inflammatory cytokine release [10,23,24,25,26]. These dual antimicrobial-immunoregulatory properties underscore the promise of *S. subspinipes*-derived peptides for both anti-infective therapy and immune support [18,25,26,27]. Transcriptomic studies suggest that *S. subspinipes mutilans* harbors additional AMP candidates, including scolopendrasin IX, which reduces inflammatory responses in rheumatoid arthritis models [8,10,28].

#### 2.2.4. Antifungal Effects

Robust antifungal activities have been reported using peptides derived from the whole body of *S. subspinipes*. A lactoferricin B-like peptide (LBLP) isolated from the whole bodies of this species potently inhibits *Candida albicans* without harming human cells [29,30]. Similarly, scolopendin from the whole body of *S. subspinipes* triggers apoptosis in *Candida albicans* through mitochondrial dysfunction, while scolopendin 2 from the whole body of *S. subspinipes* disrupts fungal membrane potentials and prompts apoptosis-like cell death through ROS-driven cellular stress and metacaspase activation [31,32]. These multi-pronged antifungal mechanisms—ranging from direct membrane disruption to programmed cell death pathways—highlight the therapeutic potential of *S. subspinipes*-derived peptides against fungal pathogens. In addition, a whole-body centipede oil extracted via n-hexane reflux demonstrated potent inhibitory effects against biofilm formation by fluconazole-resistant *Candida albicans* strains and various *Staphylococcus aureus* strains, without affecting planktonic cell growth [15].

#### 2.2.5. Antioxidant Effects

Extracts from the whole-body extract of *S. subspinipes* and their isolated compounds exert antioxidant effects by inhibiting lipid peroxidation, scavenging free radicals, and chelating metal ions [33]. Activity-guided fractionation has identified antioxidant quinoline alkaloids (e.g., 3,8-dihydroxyquinoline, also known as jineol) and phenolic compounds that demonstrate IC_50_ values comparable to synthetic antioxidants in LDL oxidation assays [33]. These substances also neutralize radical species and chelate pro-oxidant metal ions (Cu^2+^, Fe^3+^), providing multifaceted protection against oxidative stress [33].

#### 2.2.6. Antithrombotic Effects

A peptide derived from centipede venom, labeled TNGYT, exhibited inhibition of Factor Xa activity, resisting coagulation with in vitro assay and in vivo. They extend clotting times by directly inhibiting key coagulation enzymes—factor Xa and thrombin—and by curtailing fibrin polymerization [34,35]. Another novel tri-peptide labeled SQL was isolated from *S. subspinipes* whole-body extract, and showed inhibition of coagulation factors in the intrinsic blood coagulation pathway [36].

#### 2.2.7. Antitumor Effects

Scolopendrasin VII derived from whole-body *S. subspinipes* has been reported to induce necrotic cell death in leukemia cells [1]. Extracts of whole-body *S. subspinipes* display antitumor activity through direct cytotoxicity against tumor cells and indirect modulation of the host immune response. Two novel alkaloids from whole-body *S. subspinipes*, alcoholic extracts of the centipede, as well as dried powder of its whole body have been shown to suppress tumor growth in murine sarcoma and hepatoma models with minimal toxicity [37,38,39]. A purified polysaccharide–protein complex (SPPC) augments immune cell function—enhancing natural killer and T-lymphocyte cytotoxicity, shifting cytokine profiles toward Th1 responses, and lowering immunosuppressive mediators such as IL-10 and TGF-β [40]. Additionally, venom-derived peptides (e.g., scolopendrasin VII, amidated scolopin-2) induce apoptosis in cancer cells, while certain low-molecular-weight alkaloids trigger cell cycle arrest [39,41,42].

#### 2.2.8. Anti-Fibrotic Activity

Preclinical evidence suggests that *S. subspinipes* whole-body-derived alkaloids exert anti-fibrotic effects. In vitro, centipede extract significantly reduces key fibrosis markers—collagen I, fibronectin, and α-smooth muscle actin—in cultured renal cells, indicating broad inhibition of fibrogenic processes [43].

#### 2.2.9. Neuroprotective Effects

Studies also highlight the neuroprotective capacity of *S. subspinipes*. In vitro, extracts display moderate acetylcholinesterase inhibitory and copper-chelating activity, mechanisms that may attenuate cholinergic deficits and oxidative stress in neurodegenerative contexts [44]. These molecular actions protect hippocampal neurons from toxin-induced cytotoxicity, preserving cell viability [45]. Correspondingly, in vivo experiments reveal that pretreatment with *S. subspinipes* extracts mitigates neurodegeneration and seizure severity in trimethyltin-exposed mice, partly by suppressing microglial and astrocytic activation [45].

#### 2.2.10. Identification of Eligible Korean Medicine CPGs

As of April 2025, a total of 61 guidelines have been registered as developed, peer-reviewed, and accredited in compliance with the official guideline development manuals in the NCKM database, with the support of the Korean Ministry of Health and Welfare. Among these studies, seven CPGs were identified with recommendations for therapies that include *S. subspinipes* (Table 2).

Five CPGs [46,47,48,49,50] recommended herbal medicine as either a monotherapy or a combined therapy with conventional medicine. In both monotherapy and combined therapy, *S. subspinipes* was included as one of many herbal preparations for herbal medicine, of which its inclusion in herbal medicine is largely decided based on the medical decision of the doctor of Korean Medicine. Hence, recommendations with centipede as part of the varied prescription of herbal medicine were always supported by possible diagnosis that backed the inclusion of *S. subspinipes* in the herbal decoction. For example, for migraine patients to take prescriptions with centipedes, they had to have high levels of pain.

Contrary to per oral herbal medicines, subcutaneous or intramuscular acupuncture injection therapy or pharmacopuncture recommended a sole preparation of *S. subspinipes*, dried and with its head and legs removed, extracted through the aqueous process [5]. Two of these CPGs, on carpal tunnel syndrome [51] and shoulder pain [52], respectively, focused on pain relief and anti-inflammatory effects in the local area by injecting diluted extract of *S. subspinipes*.

The level of evidence for these recommendations varied from B (Moderate or Low) to C (Low or Very low), with a single guideline based on Good Practice Points/Clinical and Therapeutic Best Practices (GPP/CTB). Guidelines for herbal decoction use generally had moderate evidence levels (Level B/Moderate) for conditions such as migraine, Meniere’s disease, and gastric cancer, suggesting sufficient clinical data to support herbal decoction’s efficacy over usual care or chemotherapy. However, for conditions such as knee osteoarthritis and vertebrobasilar insufficiency-associated vertigo, the evidence was lower (Level B/Low and C/Low, respectively), reflecting limited clinical data such as non-randomized controlled trials, or potential safety concerns associated with herbal formulations containing *S. subspinipes*.

Regarding pharmacopuncture recommendations, evidence was categorized as low (Level C/Low) to very low (Level C/Very low), indicating non-randomized clinical studies, small sample sizes, and/or high potential for bias. Despite this, pharmacopuncture using *S. subspinipes* was explicitly recommended due to its observed local anti-inflammatory and analgesic effects. Clinical guidelines suggested careful selection of patients based on symptom severity, the performance of preliminary allergy tests, and administration at specific acupoints (Neiguan (PC6) and Daling (PC7)) to mitigate risks. Recommendations emphasized cautious clinical use, reflecting the balance between clinical benefit and safety considerations inherent in treatments involving *S. subspinipes*.

## 3. Discussion

The findings of this review underscore the multifaceted therapeutic potential of *Scolopendra subspinipes*-derived compounds, reflecting its historical use in traditional medicine for conditions such as chronic pain, inflammation, convulsions, and infections. TCIM has utilized centipedes extensively for their analgesic properties, primarily by topical application or ingestion of whole-body extracts. Recent scientific studies have identified that the analgesic activity of *S. subspinipes* primarily involves venom-derived peptides, notably SsmTX-I, a potent Kv2.1 ion-channel blocker. This compound produces analgesic effects comparable or superior to morphine in rodent pain models without opioid-related adverse effects. Mechanistically, it achieves pain relief through ion-channel inhibition, specifically blocking neuronal potassium channels critical to pain transmission, thereby reducing nociceptive signaling and inflammation-associated pain responses.

Investigating centipedes both as monotherapy and in combination formulations is clinically important since centipedes in traditional clinical settings are predominantly used as components of multi-ingredient herbal preparations rather than isolated therapies. Clinically used centipede formulations likely retain centipede-specific pharmacological effects, which in later studies were found to be exerted through compounds such as SsmTX-I and scolopendrasin IX, directly targeting pain and inflammatory pathways. Thus, even within complex herbal mixtures, these potent bioactive molecules contributes to understanding the current clinical practice and its observed therapeutic outcomes, indirectly supporting the traditional empirical use of *S. subspinipes*.

Beyond analgesic effects, peptides and extracts derived from centipede venom and whole-body demonstrate anti-inflammatory, antimicrobial, antitumor, antithrombotic, antioxidant, anti-fibrotic, and neuroprotective activities. These diverse effects support the traditional therapeutic roles of centipede formulations. For instance, scolopendrasin IX from centipede venom exhibits dual antimicrobial and immunomodulatory properties, activating immune responses via formyl peptide receptor 2, thereby modulating inflammatory processes central to diseases such as rheumatoid arthritis. Nevertheless, therapeutic utilization of centipede-derived substances demands caution due to documented adverse effects, including local irritation, allergic reactions, and potential immunogenic responses, necessitating stringent standardization and rigorous safety assessments.

The CPGs reviewed herein highlight how centipede-based treatments are integrated into Korean Medicine, predominantly for pain management and inflammatory conditions. Five guidelines recommended herbal decoctions containing *S. subspinipes* primarily as adjunct therapies in conditions like migraine, Meniere’s disease, and gastric cancer, supported by moderate to low-quality evidence. Notably, pharmacopuncture guidelines recommended direct injection of purified centipede extracts for carpal tunnel syndrome and shoulder pain, indicating local anti-inflammatory and analgesic efficacy despite limited clinical evidence (Level C: Low to Very Low). Guidelines emphasized careful patient selection, preliminary allergy testing, and specific administration protocols at acupoints (Neiguan [PC6], Daling [PC7]) to mitigate risks associated with centipede-derived therapies.

Despite promising preclinical outcomes, translating these findings into clinical practice requires overcoming several limitations. Current evidence remains predominantly animal-based, and human clinical data are sparse. Variability in extract preparation and composition complicates reproducibility, and mechanistic understanding of many bioactive compounds remains incomplete. Future research priorities include standardizing extraction processes, isolating key bioactive peptides, comprehensive toxicological profiling, and initiating controlled human trials. Addressing these research gaps will be essential to validating the therapeutic potential of *S. subspinipes*-derived substances and facilitating their integration into evidence-based integrative medicine.

## 4. Methods

### 4.1. Preclinical and Clinical Studies on Scolopedra subspinipes

#### 4.1.1. Search Strategy and Selection of Preclinical and Clinical Studies

We conducted a comprehensive literature review of studies investigating *S. subspinipes* (and its subspecies *S. subspinipes mutilans*) for various therapeutic effects. The keyword *Scolopedra subspinipes* was searched in PubMed, Cochrane, and EMBASE. Inclusion criteria were in vitro and in vivo experimental studies involving therapeutic compounds derived from *S. subspinipes* that are written in English, spanning biochemical assays, cell culture experiments, and animal models. The exclusion criteria were: (1) studies not in English; (2) reviews, overviews, and editorials; (3) studies focusing on the biology and biodiversity of centipedes themselves or biological analysis of its venoms; (4) studies on accidental centipede bites. This scoping review was conducted according to the PRISMA-ScR (Preferred Reporting Items for Systematic reviews and Meta-Analyses extension for Scoping Reviews) guidelines. The checklist provided by PRISMA was systematically adhered to ensure methodological rigor and transparency throughout the review process. Protocol registration was not performed as this review was primarily exploratory in nature, designed to comprehensively map the existing literature and clinical guidelines rather than answer a narrowly defined research question.

#### 4.1.2. Data Extraction from the Selected Literature

The reviewed works encompass isolated centipede venom peptides (e.g., antimicrobial peptides, ion-channel toxins), whole-body extracts prepared with different solvents (water, ethanol, etc.), polysaccharide–protein complexes purified from centipede tissue, and small molecules (such as quinoline alkaloids) derived from the centipede. In addition, reports on multi-ingredient traditional formulations containing *S. subspinipes* (for example, combined with other medicinal animals or herbs) were considered to capture the centipede’s effects in a traditional medicine context.

Across the extracted studies, a variety of experimental models were used to evaluate outcomes in each domain. For analgesic and anti-inflammatory effects, rodent models of pain (formalin-induced pain, thermal nociception, acetic acid writhing) and disease models of arthritis or neuropathic pain were employed. Antifungal and broader antimicrobial activities were tested against pathogenic microbes in culture (e.g., *Candida albicans*, *Staphylococcus aureus*) and by measuring membrane disruption or microbial viability. Antitumor effects were assessed in cancer cell lines (for cytotoxicity, cell cycle arrest, apoptosis) and in tumor-bearing mice for in vivo efficacy and immune response modulation. Antithrombotic properties were examined through clotting assays (prothrombin time, fibrinolysis tests), platelet aggregation tests, and in vivo thrombosis models in rodents. Anti-melanogenic activity was evaluated using enzymatic assays for tyrosinase and melanin production in melanocyte cell cultures. Neuroprotective effects were investigated in models of neurodegeneration, including chemical-induced hippocampal injury and transgenic mouse models of amyotrophic lateral sclerosis (ALS), as well as assays for acetylcholinesterase inhibition relevant to cognitive decline.

Key information from each study—such as the type of centipede-derived preparation, experimental model, and principal findings—was extracted and synthesized. In the Results Section above, we present a summary of therapeutic findings in each domain, emphasizing mechanistic insights and the nature of the evidence. All findings are reported objectively, and supporting references are cited. This review is intended to provide an academic overview of the multifaceted pharmacological activities of *S. subspinipes*, serving as a foundation for further research and potential translational development.

### 4.2. Clinical Practice Guidelines (CPGs) Regarding S. subspinipes

#### 4.2.1. Search Strategy and Selection of Eligible CPGs for *Scolopendra subspinipes*

To systematically identify relevant clinical practice guidelines (CPGs) that include recommendations for therapeutic applications involving *S. subspinipes*, a targeted literature search was conducted using the National Clearinghouse for Korean Medicine (NCKM) CPG database. This database is maintained and supported by the Ministry of Health and Welfare (MoHW) of the Republic of Korea and represents a reliable source for high-quality, evidence-based Korean Medicine guidelines. The search strategy aimed to capture comprehensive and contemporary CPGs specifically related to the clinical use of compounds derived from *S. subspinipes*.

The following explicit eligibility criteria were applied: (1) CPGs published up to April 2025 to ensure the most current guidelines were reviewed; (2) guidelines developed under the auspices of the Korean Ministry of Health and Welfare, to ensure a high standard of credibility and methodological rigor; (3) guidelines adhering to internationally recognized standards for guideline development, including methodologies established by the Cochrane network, the Grading of Recommendation, Assessment, Development, and Evaluation (GRADE) system, and the Appraisal of Guidelines, Research, and Evaluation (AGREE) tool; and (4) guidelines that explicitly recommended therapeutic applications of compounds or extracts derived from *S. subspinipes*. Guidelines recommending these applications either as standalone treatments (monotherapy) or in combination with other therapies (polytherapy) were considered eligible.

#### 4.2.2. Basic Elements of the CPGs, Their Recommendations, and Evidence Appraisal

The data extraction encompassed identification of the targeted conditions or diseases, classified according to the Korean Classification of Diseases, as well as the specified target populations (where available). Details on the specific therapeutic interventions involving *S. subspinipes* were systematically recorded, specifying whether they were recommended as monotherapy or polytherapy. Detailed prescriptions were recorded when available. Information regarding the evidence base supporting these recommendations was also extracted.

The extracted data also encompassed the grade of recommendations along with the associated rationale, facilitating an understanding of the decision-making context behind guideline recommendations. Recommendations were classified as Grade A to D depending on their recommendation level for clinical contexts considering their potential therapeutic benefits and risks. Furthermore, as stipulated in the guideline manuals, the recommendations took into consideration broader clinical decision-making factors such as clinical availability, treatment cost-effectiveness, patient preferences, expert consensus, survey results regarding clinical practice utilization, inclusion in authoritative resources (e.g., pharmacopeias, textbooks), and official data from authorized organizations. These factors were systematically reviewed and extracted from the included CPGs.

Decisions and recommendations made in the CPGs regarding the quality and level of evidence supporting the efficacy and safety of therapeutic applications involving *S. subspinipes* were also extracted. According to the guidelines’ manuals, evidence quality was categorized into four levels: “High”, “Moderate”, “Low”, and “Very Low”.

## Figures and Tables

**Figure 1 toxins-17-00229-f001:**
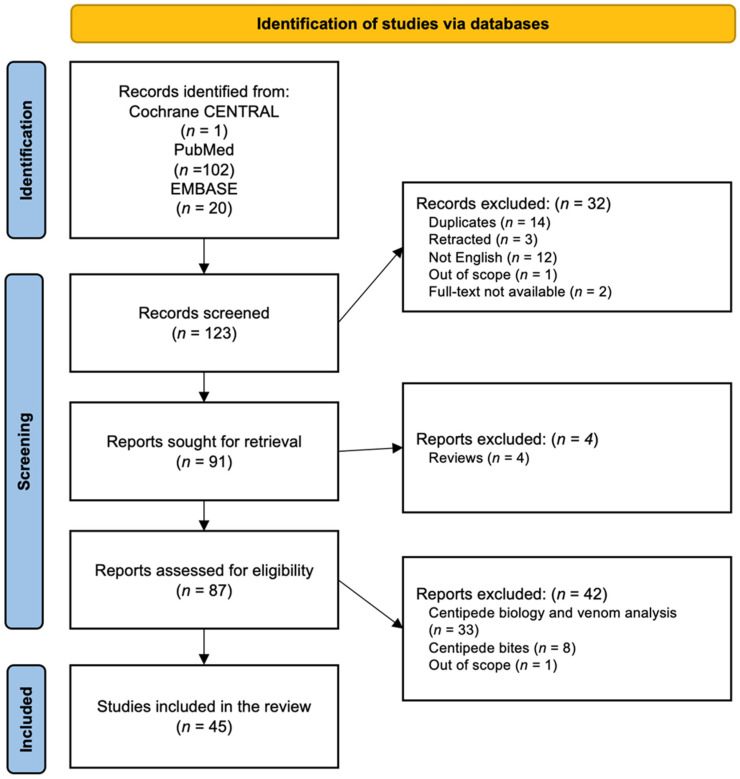
Flowchart.

**Table 1 toxins-17-00229-t001:** Summary of the effects of *Scolopendra subspinipes*.

Therapeutic Effect	Key Findings	Mechanisms	Key Compounds (Chemical Class, Source)	References
Analgesic	Reduces inflammatory, acute, and neuropathic pain via ion channel blockade and adrenergic pathways.	Kv2.1 ion-channel inhibition, α2-adrenergic receptor activation, NF-κB suppression.	SsmTX-I (peptide, centipede venom). Centipede pharmacopuncture (purified aqueous extract, whole-body dried centipede). Zhi Jing San (herbal medicine, whole-body dried centipede).	[3,4,5]
Anti-Inflammatory	Suppresses NF-κB signaling, proinflammatory cytokines, and oxidative mediators in systemic and localized models.	NF-κB inhibition, cytokine suppression, iNOS/COX-2 downregulation.	Scolopendrasin X (peptide, centipede venom). Zhi Jing San (herbal medicine, whole-body dried centipede). Soufeng sanjie formula (herbal medicine, whole-body dried centipede).	[3,6,7,8,9,10,11,12]
Antimicrobial	AMPs target bacteria/fungi via membrane disruption or immune modulation; transcriptomics reveal new candidates.	AMPs, FPR2 activation, ROS-mediated apoptosis, protein synthesis inhibition.	Scolopin 1 (peptide, centipede venom). scolopendrasins III, V, VII, IX, and X (peptide, centipede venom).	[8,10,13,14,15,16,17,18,19,20,21,22,23,24,25,26,27,28]
Antifungal	Peptides disrupt *Candida* membranes or induce apoptosis via ROS, Ca^2^⁺ dysregulation, and mitochondrial damage.	Membrane permeabilization, ROS generation, apoptosis induction.	LBLP: lactoferricin B-like peptide (peptide, whole-body centipede). Scolopendin (peptide, whole-body centipede). Scolopendin 2 (peptide, whole-body centipede).	[15,29,30,31,32]
Antioxidant	Compounds scavenge free radicals, chelate Cu^2^⁺, and inhibit LDL oxidation.	Radical scavenging, metal ion chelation, antioxidant assays.	Compound 1: 3,8-dihydroxyquinoline (alkaloid, whole-body centipede). Compound 2: 2,4-di-tert-butylphenol (phenolic compound, whole-body centipede). Compound 3: 2,8-dihydroxy-3,4-dimethoxyquinoline (alkaloid, whole-body centipede).	[33]
Antithrombotic	Inhibits thrombin, FXa, and platelet activation; reduces thrombus formation in vivo.	Coagulation factor inhibition, platelet signaling disruption.	TNGYT (peptide, centipede venom). Compound 1 (acyclated polyamines, whole-body centipede). Compound 2 (sulfated analogue of jineol, whole-body centipede). Compound 3 (quinolone alkaloid, whole-body centipede). Compound 4 (indole acetic acid, whole-body centipede). Tri-peptide labeled SQL (peptide, whole-body centipede).	[34,35,36]
Antitumor	Induces apoptosis, modulates immunity, suppresses angiogenesis, reduces tumor burden in models.	Cell cycle arrest, TRAIL pathway, SPPC-mediated immune shift.	Scolopendrasin VII (peptide, centipede venom). Compound 1–2 (isoquinolone alkaloids, whole-body centipede). Scolopentide (peptide, whole-body centipede). Polysaccharide–protein complex (compound, whole-body centipede). Ethanolc extracts, and dried powder of whole-body centipede.	[1,37,38,39,40,41,42]
Anti-Fibrotic	Suppresses fibrosis markers (collagen I, fibronectin, α-SMA) in renal cell models.	Inhibition of fibrotic protein expression.	Scolopenolines A-L (alkaloids, whole-body centipede).	[43]
Neuroprotective	Inhibits AChE, chelates Cu^2+^, reduces neuroinflammation, preserves neuronal viability in vivo.	AChE inhibition, anti-inflammatory, antioxidative, neuroprotection.	Water extract of whole-body centipede.	[44,45]

Kv2.1: Voltage-gated potassium channel subfamily 2 member 1; NF-κB: Nuclear factor kappa-light-chain-enhancer of activated B cells; iNOS: Inducible nitric oxide synthase; COX-2: Cyclooxygenase-2; AMPs: Antimicrobial peptides; FPR2: Formyl peptide receptor 2; ROS: Reactive oxygen species; LBLP: Lactoferricin B-like peptide; LDL: Low-density lipoprotein; FXa: Coagulation factor Xa; TNGYT: Thr–Asn–Gly–Tyr–Thr peptide sequence; SQL: Ser–Gln–Leu tripeptide sequence; TRAIL: TNF-related apoptosis-inducing ligand; SPPC: Polysaccharide–protein complex.

**Table 2 toxins-17-00229-t002:** Clinical practice guidelines with mentions of the effect of *Scolopendra subspinipes*.

CPG	Specific Conditions	Level of Evidence	Monotherapy or Combined Therapy	Comparison	Details	Origin of Primary Study
Recommendations on herbal decoction including *S. subspinipes*						
Knee Osteoarthritis [46]	Liver-kidney deficiency syndrome	B/Low	Monotherapy	Placebo	Yongbyeol capsule includes potentially toxic herbs like centipede (*S. subspinipes*). Caution regarding adverse effects is necessary, and safety should be monitored through clinical pathology testing with long-term use.	China (1)
Migraine [47]		B/Moderate	Monotherapy	Usual care (Flunarizine)	Herbal medicine may be more effective than conventional medication (Flunarizine) for symptom improvement in migraine patients.	China (1)
				Usual care (pharmacological interventions)	Herbal treatment with Tonggyu-Hwalhyeol-Tang modifications (which include *S. subspinipes*) can reduce migraine symptoms, attack frequency, and duration compared to standard pharmacological care.	China (1)
Dizziness (vertigo) [48]	Vertebrobasilar insufficiency-associated vertigo	C/Low	Combined therapy	Usual care (nimodipine)	Combined therapy with herbal medicine and standard anti-vertigo medications (vascular/circulatory enhancers) can be considered over medication alone.	China (1)
					(Banxia-Baizhu-Tianma decoction modified using 2 pcs of dried *S. subspinipes*.)	
	Meniere’s disease	B/Moderate	Combined therapy	Usual care (anti-vertigo medication)	Combined therapy with herbal medicine and standard anti-vertigo medication is recommended over anti-vertigo medication alone.	China (1)
					Jingxuan decoction modified with 6 g of *S. subspinipes*.	
Gastric cancer [49]	Herbal medicine combined with adjuvant chemotherapy	B/Moderate	Combined therapy	Chemotherapy	For patients undergoing radical gastrectomy followed by adjuvant chemotherapy, combined treatment with herbal medicine containing *S. subspinipes* may enhance chemotherapy effectiveness and should be considered.	China (1)
Autism [50]	Hyperactivity, tic disorder	GPP/CTB	Combined therapy	Behavioral/educational therapy	Based on expert consensus, the addition of Chaihu-Longgu-Muli decoction is recommended for symptom improvement in children with autism spectrum disorder presenting as liver qi stagnation syndrome. In cases of hyperactivity or tic disorders, additionally include “... *S. subspinipes* (5 g) … [50]”	China (1)
Recommendations on *S. subspinipes* pharmacopuncture						
Carpal Tunnel Syndrome [51]		C/Very low	Combined therapy	Electroacupuncture	**Clinical considerations:** Based on the literature, pharmacopuncture for peripheral nerve conditions commonly uses “… *S. subspinipes* pharmacopuncture … based on patient status. … Allergy skin tests should precede bee venom or centipede pharmacopuncture. Location: Neiguan (PC6) and Daling (PC7) points, and electroacupuncture may be administered concurrently [51]”.	Korea (1)
Shoulder Pain [52]	Extreme pain	C/Low	Monotherapy	Acupuncture	Pharmacopuncture using *S. subspinipes* “… may be considered for adult patients whose primary symptom is shoulder pain [52]”.	Korea (1)

CPG: clinical practice guideline; GPP/CTB: Good Practice Points/Clinical and Therapeutic Best Practices.

## Data Availability

The original contributions presented in this study are included in this article and Appendix A. Further inquiries can be directed to the corresponding author.

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
