# Peer review of "Therapeutic Potential of Scolopendra subspinipes: A Comprehensive Scoping Review of Its Bioactive Compounds, Preclinical Pharmacology, and Clinical Applications"

_toxins, 2025, doi:10.3390/toxins17050229_

Round 1
Reviewer 1 Report
Comments and Suggestions for Authors
Review: Therapeutic Potential of Scolopendra subspinipes: A Comprehensive Review of Its Bioactive Compounds and Preclinical Pharmacology
General: Make sure that all scientific name were written in italic such as Scolopendra subspinipes and S. subspinipes etc.
Abstract:
- Information of Scolopendra subspinipes in therapeutic aspects should be more clarified in term of active compounds. Were all of them extracted from venom or other parts? If all mentioned compounds were from venom, indication of venom must be presented in the title.
Introduction:
- Line 37 the term “S. subspinipes-derived substances” were they extracted from venom. Authors should clarify the origin of each substance.
- Line 44 the term “S. subspinipes’s bioactivities” makes the readers wondering these bioactivities are from venom or which paths please specific.
- Suggestion: Toxic outcomes following bitten by centipede should be mentioned in order to better understanding the therapeutic effects of these compounds.
Method:
- Indication of inclusion and exclusion criteria must be clearly clarified as the author also mention the inclusion criteria in result section in line 89.
Result:
- Line 89 author mentioned that “42 studies were further excluded because they did not meet the inclusion criteria”. Please clearly mentioned why those works did not meet the criteria.
- Table 1: It should be more informative if author can separate which compound was from venom which one from other part or whole body of centipede. Indication of peptide or compound must be shown in this table.
- Indication of abbreviation must be shown in end of the table.
- Indication of novel compounds including their mechanism of action must be included in the table1.
- Line 165, Candida albicans must be italic.
- Line 170 albicans must be italic.
- Make sure that all hyphen points are using properly.
- Line 191, please indicate how the subspinipes–derived compounds (e.g., compounds 1, 2, and 4) were extracted from which parts of centipede.
- Line 203-204 author mentioned “The centipede S. subspinipes displays antitumor activity through direct cytotoxicity 203 against tumor cells and indirect modulation of the host immune response”. My understanding it means the whole body of centipede exhibited antitumor activity. Interpretation about other substances in centipede body should be mentioned.
Discussion:
Information regarding the use of centipede in traditional treatment can be more mentioned as well as the venom compounds in centipede venom in order to associate with the therapeutic mechanism especially in analgesic mechanism. Detail and knowledge regarding pain pathway can be more included. Moreover, limitation of using the substances from centipede must be mentioned including adverse effects.
Comments on the Quality of English Language- Please the scientific name writing which point which need to be italic or hyphenation.
Reviewer 2 Report
Comments and Suggestions for Authors
In this review, the authors demonstrate the multifaceted therapeutic potential of compounds derived from Scolopendra subspinipes. They point out the difficulties of clinical application of centipede extracts, which are still in the preclinical trial stage despite their activity, and provide a perspective on future developments by referencing their long-standing use in traditional Chinese and Korean medicine. The authors also highly praised the method of organizing the literature and providing a concrete method for preparing a review like this one. We believe that the following minor corrections will help to broaden the reader's understanding.
- The following characters in Table 1 are illegible. Please correct each one so that they are legible.
CE±2, NF-CE∫B, Ca¬≤‚Å∫, Cu¬≤‚Å∫,
Reviewer 3 Report
Comments and Suggestions for Authors
The authors present a review of the therapeutic potential of the venom or whole organism (ground up without refinement or subjected to extraction) preparations derived from the Chinese red-headed centipede (Scolopendra subspinipes). Use of this centipede in one manner or another has been part of Chinese medicine historically, and the authors would like to see it advanced as therapy in medicine at large by defining mechanism and advancing to clinical trials. This organism has great potential as a therapeutic source, which is the rationale for this review article. I have several comments.
Comments
- Methods: “In addition, reports on multi-ingredient traditional formulations containing S. subspinipes (for example, combined with other medicinal animals or herbs) were considered to capture the centipede’s effects in a traditional medicine context.”
These works need to be excluded – the authors have no way of knowing if the combination of the centipede’s components with the other parts of the mixture or the other components are responsible for the effects seen. I would be curious to see how many of the papers used only had the centipede as the item tested.
- Results. The authors claim that this centipede as a whole or extract is responsible for analgesic, anti-inflammatory, antimicrobial, antifungal, antioxidant, antithrombotic, antitumor, anti-fibrotic, and neuroprotective effects in in vitro, cellular, and whole organism models. There is no attempt to sort out what configuration of the centipede (e.g., venom, ground up centipede powder), what testing system, and what mechanisms responsible are associated with one another in Table 1. There are no figures and no tables that provide more granular information that would be critical to the readership.
Table 1 has a variety of characters in it I have not seen before. They include: NF-OE∫B, Ca¬≤‚Å∫, Cu¬≤‚Å∫, and OE±-SMA. The authors need to clarify what they mean.
The subsections discussing the various effects exerted by the centipede (e.g., anti-inflammatory) add some detail, but the reader is left with a diffuse, “it cures everything” impression from the authors.
Given the lack of exclusion of studies that combine the centipede with other components, lack of detail granularity, and lack of diagrams/focus, the work does not provide the readership with a reasonable understanding of the space at this time and plausible future directions of research.
Minor comments
- Please use italics for the species names.
Round 2
Reviewer 1 Report
Comments and Suggestions for Authors
Authors have made the correction based on my comments. This revised version is ready for publication.
Reviewer 3 Report
Comments and Suggestions for Authors
The authors have substantially improved their manuscript. When appropriately couched, the assertation that materials used from the centipede in traditional compounded medicines lends relevance, albeit at the expense of not being to sort out the singular contribution of the organism. At least the parts and extracts of the organism are identified with the relevant medicines and potential cures.